# NanoBiT System and Hydrofurimazine for Optimized Detection of Viral Infection in Mice—A Novel in Vivo Imaging Platform

**DOI:** 10.3390/ijms21165863

**Published:** 2020-08-15

**Authors:** Natasa Gaspar, Giorgia Zambito, Iris J. C. Dautzenberg, Steve J. Cramer, Rob C. Hoeben, Clemens Lowik, Joel R. Walker, Thomas A. Kirkland, Thomas P. Smith, Wytske M. van Weerden, Jeroen de Vrij, Laura Mezzanotte

**Affiliations:** 1Department of Radiology and Nuclear Medicine, Erasmus Medical Center, 3015 GD Rotterdam, The Netherlands; n.gaspar@erasmusmc.nl (N.G.); g.zambito@erasmusmc.nl (G.Z.); c.lowik@erasmusmc.nl (C.L.); 2Erasmus Medical Center, Department of Molecular Genetics, 3015 GD Rotterdam, The Netherlands; 3Percuros B.V., 2333 CL Leiden, The Netherlands; 4Medres Medical Research GMBH, 50931 Cologne, Germany; 5Leiden University Medical Center, Department of Cell and Chemical Biology, 2333 Leiden, The Netherlands; I.J.C.Dautzenberg@lumc.nl (I.J.C.D.); s.j.cramer@lumc.nl (S.J.C.); r.c.hoeben@lumc.nl (R.C.H.); 6Department of Oncology CHUV, UNIL and Ludwig Cancer Center, 1011 Lausanne, Switzerland; 7Promega Biosciences L.L.C., San Luis Obispo, CA 93401, USA; Joel.Walker@promega.com (J.R.W.); thomas.kirkland@promega.com (T.A.K.); thomas.smith@promega.com (T.P.S.); 8Erasmus Medical Center, Department of Urology, 3015 GD Rotterdam, The Netherlands; w.vanweerden@erasmusmc.nl; 9Erasmus Medical Center, Department of Neurosurgery, 3015 GD Rotterdam, The Netherlands; j.devrij@erasmusmc.nl

**Keywords:** nanobit system, hibit tag, hydrofurimazine, oncolytic virus, bioluminescence imaging

## Abstract

Reporter genes are used to visualize intracellular biological phenomena, including viral infection. Here we demonstrate bioluminescent imaging of viral infection using the NanoBiT system in combination with intraperitoneal injection of a furimazine analogue, hydrofurimazine. This recently developed substrate has enhanced aqueous solubility allowing delivery of higher doses for in vivo imaging. The small high-affinity peptide tag (HiBiT), which is only 11 amino-acids in length, was engineered into a clinically used oncolytic adenovirus, and the complementary large protein (LgBiT) was constitutively expressed in tumor cells. Infection of the LgBiT expressing cells with the HiBiT oncolytic virus will reconstitute NanoLuc in the cytosol of the cell, providing strong bioluminescence upon treatment with substrate. This new bioluminescent system served as an early stage quantitative viral transduction reporter in vitro and also in vivo in mice, for longitudinal monitoring of oncolytic viral persistence in infected tumor cells. This platform provides novel opportunities for studying the biology of viruses in animal models.

## 1. Introduction 

The development of novel strategies for pre-clinical and clinical imaging of viral replication and dissemination is of crucial importance [1]. An important tool used to deconvolute the pathways of viral infection are reporter viruses, which are constructed to study the course of infection. Bioluminescent imaging (BLI) is a highly effective imaging modality used in small animal models enabling longitudinal in vivo analysis of viral pathogenesis and determination of the efficacy of therapeutic interventions due to its high sensitivity and high signal to background ratio. BLI relies on luciferase-based gene reporters to detect sites of viral infection and to quantify viral transduction by correlation with in vivo photon emission within a single animal model [2].

A recombinant virus genetically engineered to express a luciferase produces light from infected cells after substrate (luciferin) administration, which can be detected using a sensitive, charged-coupled device (CCD) camera. This approach has been used in animal models for several viruses, e.g., Dengue virus, Herpes simplex virus 1 (HSV-1), Sindbis virus, Vaccina virus, Influenza virus and oncolytic viruses (OVs) [3,4,5,6,7,8,9].

Generating recombinant reporter viruses remains a challenge. The smaller the inserted reporter gene, the more stable the viral genome remains, making small reporter tags applicable to different viral strains that do not tolerate large genome alterations such as RNA viruses, or being near the limit of their packaging capacity (e.g., armed oncolytic DNA viruses) [10]. Our proposed solution to this problem is the use of a small, multifunctional reporter tag that can be inserted anywhere in the viral genome without substantially altering its properties. HiBiT is a split-reporter tag derived from NanoLuc binary technology (NanoBiT) [9] with high affinity for the split-LgBiT reporter. The NanoBiT system is highly suitable for this purpose. NanoLuc (Nluc) luciferase is a relatively small (19 kDA), ATP independent bioluminescent enzyme and is significantly brighter than Firefly (Fluc) and Renilla (Rluc) luciferases [11].

In our system the very small HiBiT tag (33 base pairs/11 amino-acids) is inserted into the viral genome and interacts with the complementary large part (LgBiT) expressed in the infected cells upon infection of those cells with the tagged virus, reconstituting the full Nluc luciferase. Recently, a Flavivirus reporter has been generated using the NanoBiT complementation approach [7]. Tomokazu et al. demonstrated its utility for in vitro screening of active compounds and ex vivo measurement of viral persistence. However, with that system, in vivo imaging was not possible [7,11]. In fact, while the small size of NanoLuc is an asset as a reporter in viral imaging in small animals, a major limitation has been the poor solubility and pharmacokinetics of its substrate, furimazine, in biological environments. This limits the amount of the substrate that can be injected in one dose and makes intravenous injection the preferred route of administration in order to reach adequate in vivo imaging sensitivity [8]. To overcome the in vivo limitations of furimazine, we used hydrofurimazine (HFz), along with a formulation enabling fast reconstitution. This allowed us to dissolve higher amounts of substrate for intraperitoneal (i.p.) injection in small animals, achieving optimal sensitivity [9,12,13].

OVs are gaining momentum as an emerging therapy for cancer treatment using engineered viruses to selectively kill tumor cells. Improving in vivo imaging of oncolytic viruses would improve safety and efficacy assessments and localization of viral biotherapeutics, with the potential to translate those findings towards the clinic [1]. A new generation of imaging tools is expected to improve in vivo analysis over current methods, which are often not suitable for imaging the therapeutic efficacy of particular types of OVs.

Here we demonstrate the power of our system for viral tracking over the course of 6 weeks post infection using an oncolytic adenovirus as model virus. This resulted in the HAdV-5-DELTA-24-RGD-GFP-T2A-HiBiT oncolytic adenovirus carrying the small HiBiT tag. The adenoviral infection and further replication were monitored via expression of HiBiT tag from the E3 promoter which is activated by the E1A proteins and expressed early after infection [14] and later reconstitution with the LgBiT protein expressed in the cytosol of PC-3 cells.

The approaches established within this study will have a direct application to various studies where sensitive detection is needed for a better understanding of infection dynamics in vivo, making it a preferable detection method especially for viruses not tolerating accommodation of large inserts or are at the limit of their packaging ability.

## 2. Results

### 2.1. Expression of HiBiT and Preserved Virulence of the Reporter Virus In Vitro

We performed a lentiviral transduction of prostate cancer cell lines PC-3, PC346C and PC339C to induce expression of LgBiT. PC-3 and PC346C are adherent cell lines while PC339C grows in suspension. In order to determine the functionality and the expression stability of the HiBiT tag from the NanoBiT system, a series of cell lines were infected with the oncolytic reporter virus (HAdV-5-DELTA-24-RGD-GFP-T2A-HiBiT). After the NanoBiT complex was reconstituted successfully, it formed a functional NanoLuc enzyme after infection with the virus (Figure 1A) over several dilutions of the initial viral stock (Figure 1B and Appendix A). The signal was clearly visible at 24 h post infection with all prostate cancer cell lines used, including the PDX-derived cell lines (Figure 1B and Appendix A). Moreover, the bioluminescent signal was specific and it quantitatively correlated with viral infectivity (Figure 1D and Appendix A). Furthermore, when we performed the assay with addition of intravenous immunoglobulins (IvIg) in order to neutralize the HiBiT-reporter virus (by adding IvIg) we observed 2- to 3-fold drop in luminescence output (Figure 1D and Appendix A). We have calculated an inter-assay variability of 16% and an intra-assay variability of 80%, parameters that are highly influenced by stochastic variability in virus-cell interactions. The reporter virus contains a GFP-T2A-HiBiT-expression cassette under the control of a viral E3 promoter where expression of the cassette is only initiated after efficacious viral replication. Cells from this assay (Figure 1B,C) were imaged to visualize the presence of virus based on the amount of GFP positive cells. The GFP signal could be detected at multiple viral dilutions ranging from a multiplicity of infection (MOI) of 4 to 0.16 (Figure 1E).

### 2.2. Sensitive Longitudinal In Vivo monitoring of HiBiT-Reporter Oncolytic Virus 

We next evaluated the in vivo efficacy of the HiBiT-reporter virus to monitor viral infection dynamics over time by stable expression of the HiBiT tag from the reporter-virus and successful infection of PC-3-LgBiT expressing cells. Three weeks after implantation of tumor cells, mice were infected intratumorally with the HiBiT-reporter virus (*n* = 8) or with Phosphate Buffered Saline (PBS) (*n* = 4) (control). The HFz substrate in its novel formulation proved particularly well-suited for these experiments, where a broad imaging time window was required. In fact, the substrate showed a stable signal 15 min after intraperitoneal administration when formulated in P-407 in mice carrying NanoLuc expressing cells [13]. As early as 24 h post viral injection, all mice showed specific photon emission, suggesting successful infection of PC-3-LgBiT tumor cells in vivo and good distribution of HFz to the tumor site after i.p. injection, leading to sustained bioluminescent emission by the reconstituted NanoLuc luciferase (Figure 2B,C).

On the contrary, when furimazine was injected intraperitoneally (4 mg/kg) we could not detect any signal (data not shown). When imaging was repeated at day 10 post infection (dpi), most tumors still displayed photon emission, after substrate administration, suggesting the presence of the virus in these tumors (Figure 2C). In six out of nine mice the photon emission increased by 1.5-fold, and this group was designated the ‘high signal BLI group’ (Figure 2B,C). In the other three mice the light emission from the infected tumors decreased, and we referred to this group as the ‘low signal BLI group’ (Figure 2C,D). When mice were imaged on day 20 post viral administration, photon emission of the ‘low BLI signal group’ tumors was indistinguishable from the background levels (Figure 2C,D) measured in the control group with an average signal of 3.5 × 10^3^ photons per second (ph/s) (Figure 2A,D). The ‘high BLI signal group’ showed a continuous increase in photon emission; on day 43 post viral administration, one mouse displayed a signal of 1.3 × 10^5^ ph/s, 30 times higher than the average signal of the control group (Figure 2B,D) suggesting the persistence of HiBiT-reporter virus for 6 weeks post administration.

As for the tumor growth kinetics shown in Figure 2E, we observed that PC-3-LgBiT tumors treated with the HiBiT-reporter virus had a reduced tumor growth rate compared to the PBS control group (Figure 2E). The oncolytic virus intervention did not eradicate the tumor completely. 

To confirm the presence of the reporter virus in tumor tissue and the specificity of the BLI signal, tumor sections (obtained 43 dpi for the treated mice and 20 dpi for the control mice) were analyzed for adenoviral capsid proteins. While staining was absent in control tumor sections injected with PBS (Figure 3A), the virus-treated tumors displayed a clearly visible signal (Figure 3B), indicating the presence of virus.

## 3. Discussion

Bioluminescent reporters have become routine tools for monitoring viral spread in living animals through sequential imaging [2,4]. Moreover, the ability to detect viral infection at the earliest possible time point and follow it up longitudinally over time in each animal provides valuable data about viral infection pathways as well as the progression of infection. Unfortunately, the large size of many luciferase reporter genes may alter the virulence and replication of viruses, which makes these genes unsuitable for introduction into most viral genomes, particularly those which do not tolerate incorporation of large transgenes. Other issues with recombinant viruses include reversion to unmodified virus (e.g., short single strand RNA viruses) and insufficient loading capacity (e.g., armed DNA or RNA oncolytic viruses), all of these viruses could benefit from an even smaller reporter in order to image them in vivo. For instance, the first in vivo BLI tracking of Influenza virus [15,16] was achieved using the very small and bright NanoLuc luciferase (19 kDa), allowing sensitive in vivo imaging of the recombinant virus in mice and ferrets.

Using the NanoBiT system, we engineered an oncolytic replication-competent HiBiT-reporter adenovirus harboring the smallest reporter tag reported for BLI imaging. In our initial in vitro experiments, we infected different LgBiT expressing cell lines with the HiBiT-reporter virus. Twenty-four hours post infection we could clearly detect a signal resulting from the successful viral transduction and NanoBiT system reconstitution. Furthermore, when we added a neutralizing antibody, we observed a 2- to 3-fold drop in the BLI signal as a result of antibody interference with viral uptake in cells. This demonstrates that our novel bioluminescent system can be used for screening of potential antivirals or for the screening of oncolytic in patients since oncolytic viruses also induce strong anti-tumor immune responses, which may act against all tumor cells. To further address this point, we plan to investigate the use of the NanoBiT system in the context of oncolytic virus performance in an immune-competent syngeneic mouse model.

We anticipate that pre-clinical research into (oncolytic) viral infections will strongly benefit from this non-invasive bioluminescent viral screening tool, enabling fast and simple longitudinal readout protocols applicable for rapid drug screening and viral-risk virus infectivity and transduction in a range of cell lines. To enable in vivo imaging of the infection course with our HiBiT-reporter virus, we employed hydrofurimazine, a substrate with improved solubility compared to furimazine [13]. The substrate was formulated this in a non-toxic, highly water-soluble excipient, which allows much higher substrate loading and extended light emission in vivo after a single substrate injection. This was necessary since these experiments required biosafety level 2 protocols and were carried out under containment in an imaging box. The animals were injected with the substrate and then transported to the animal imaging facility.

Having generated these tools, we were able to monitor infection dynamics of the HiBiT-reporter virus in living mice using established PC-3-LgBiT tumors over 6 weeks with hydrofurimazine. We detected infectious virus production as early as 24 h post viral administration and as late as 43 days post infection. These results suggest efficacious viral infection and successful in vivo complementation of the two NanoBiT subunits (LgBiT and HiBiT) within the transgenic xenografts.

The variation between the high and the low BLI signal we observed in this murine model is representative of the biological variation observed in clinical trials with oncolytic (adeno) viruses [16]. Using our imaging system, it was also evident that the oncolytic virus did not reach the entire tumor mass, in line with the performance of oncolytic adenoviruses evaluated in clinical trials. This could be the result of a number of different factors, e.g., presence of viral-limiting supportive tissue or extracellular matrix and/or lack of the viral CAR receptor (coxsackievirus and adenovirus receptor) on many tumor cells [17,18]. Still, the incomplete spread of oncolytic viruses to all tumor cells does not necessarily prevent complete tumor clearance in an actual human assessment [19]. This technology is applicable to study a wide range of viruses, and we predict that it will be particularly useful for viruses that do not tolerate larger transgene insertions.

## 4. Materials and Methods

### 4.1. Cell Lines and Cell Culture Conditions

Human prostate cancer cell lines were used as a model system in this study. The PC-3 cell line was cultured in Roswell Park Memorial Institute (RPMI) 1640 Medium (Sigma, St. Louis, MO, USA) supplemented with 10% of FBS and 1% Penicillin-Streptomycin. The cells were kept at 37 °C in a humidified atmosphere containing 5% CO_2_. Human embryonic kidney (HEK293T) cells were grown in Dulbecco’s modified Eagle’s medium DMEM (Sigma, St. Louis, MO, USA), supplemented with 10% fetal bovine serum and 1% penicillin/streptomycin.

PC339C and PC346C cell lines are established from patient-derived xenografts and were chosen as bench mark models because of their known transduction potential [20]. PC339C and PC346C cell lines were cultured in prostate growth medium (PGM) as described previously [20]. In short, culture medium DMEM-F12 (Cambrex, BioWhittaker, Verviers, Belgium) was supplemented with 2% FCS (PAN Biotech, Aidenbach, Germany), 1% insulin-transferrin-selenium (GIBCO BRL, Gaithersburg, MD), 0.01% BSA (Boehringer-Mannheim, Germany), 10 ng/mL epidermal growth factor (Sigma-Aldrich, Milan, Italy) and penicillin/streptomycin antibiotics (100 U/mL penicillin, 100 µg/mL streptomycin; (Cambrex, BioWhittaker, Verviers, Belgium). Cell lines were passaged when they reached 80% confluence (80% medium of T-175 flask covered by cell aggregates).

### 4.2. Lentiviral Vector Construction and Cell Transduction

Vector production and cell transduction were performed under appropriate biosafety level conditions (ML-II) in accordance with the National Biosafety Guidelines and Regulations for Research on Genetically Modified Organisms (GMO permit 99-163 from the Bureau of genetically modified organisms, The Netherlands). Procedures and protocols were reviewed and approved by the EMC Biosafety Committee. To create the pCDH-EF1-ATG-LgBiT-T2A-copGFP lentiviral vector, we first excised the 1929 luciferase gene from the vector pCDH-EF1-ATG-1929-T2A-copGFP (Promega, Madison, WI, USA) with restriction enzymes BamHI and NotI in NEB buffer 3.1. from New England BioLabs.

The LgBiT gene was amplified from the vector pBIT1.3.-C[LgBiT] (Promega) using the following primers; forward primer with a NotI restriction site 5′- TTT GCGGCCCGCATGGTTACTCGGAAC-3′ and 3′-GGATCCATGCTGGCTCGAGCGGTGG-5′ with a BamHI restriction site. The amplified PCR product (LgBiT gene) was cloned in the above-prepared pCDH-EF1-ATG-T2A-copGFP recipient vector to create the pCDH-EF1-ATG-LgBiT-T2A-copGFP vector using the NotI and BamHI restriction sites.

A self-inactivating lentivirus pCDH-EF1-LgBiT was produced by transfection of HEK293T packaging cells by transient transfection of HEK293T cells with three packaging plasmids pCMV-VSVG, pMDLg-RRE, pRSV-REV (Addgene, Cambridge, MA, USA) and PEI transfection reagent 1 mg/mL per μg DNA. Procedures were previously described in detail [21]. Lentiviral supernatant was collected after 48 and 72 hours and filtered (0.45 µm). Subsequent quantification of virus was performed using a standard antigen capture HIV p24 ELISA (ZeptoMetrix, Buffalo, NY, USA). PC-3, PC346C and PC339C cells were grown in culture dishes to 50% confluence in culture medium and were infected with the lenti-viral stock, resulting in LgBiT expression. Cells were transduced with MOI 1 particle per cell of pCDH-EF1-LgBiT-T2A-copGFP, lentivirus in the presence of polybrine (hexametride bromide, Sigma-Aldrich) at a final concentration of 8 µg/mL. Stable clones were selected via the limited dilution method.

HEK293T cells were seeded in a T-25 flask at a density of 5 × 10^5^ cells and transduced with either EF1-NanoLuc-T2A-copGFP lentivirus plus polybrene (hexametridine bromide) (Sigma, St. Louis, MO, USA) at a final concentration of 8 μg/mL. Cells were sorted for GFP expression using FACS (BD-FACS AriaIII, BD Biosciences). Transgene expression was confirmed by the presence of the green fluorescent protein copGFP from (excitation/emission maximum = 475/509 nm).

### 4.3. Reporter Virus Construction

The oncolytic HAdV-5-DELTA-24-RGD-GFP-T2A-HiBiT virus was constructed and produced according to previously described protocols [22]. The adenovirus has a 24-base pair deletion in the viral E1A gene, which disrupts the retinoblastoma protein (Rb)-binding capacity of this protein and facilitates selective replication in tumor cells with a dysfunctional Rb-pathway. The RGD peptide in the fiber protein allows the virus to bind and enter cells through cell surface integrins ανβ3/5, which are often overexpressed on the surface of cancer cells. The virus contains the eGFP-T2A-HiBiT-expression cassette under control of the viral E3 promoter. Expression only occurs when viral replication is initiated. The 2A-HiBiT sequence was synthesized with BsrGI restriction site overhangs (TGTACAAGGCTGAGGGCAGAGGAAGTCTTCTAACATGCGGTGACGTGGAGGAGAATCCCGGCCCTGTGAGCGGCTGGCGGCTGTTCAAGAAGATTAGCTAATGTACA) and was subsequently cloned in the plasmid pShuttle-ΔE3-Fib-RGD-ADP-EGFP-Kana. The GFP-2A-HiBiT containing fragment (after PacI+AatII+ScaI digestion) was recombined with SpeI-linearized pAdEasy-1 resulting in the plasmid pAE-RGD-GFP-2A-HiBiT. Using a standard recombination procedure [22] this plasmid was recombined with pSh+pIXresulting in HAdV-5-DELTA-24-RGD-GFP-T2A-HiBiT. The virus was rescued in HER911 cells [23]. To prevent heterologous recombination with the viral E1 sequence present in the HER911 genome, upscaling of the virus was performed in A549 cells. The virus was isolated by double cesium-chloride density gradient purification and a plaque assay on HER911 cells was performed to determine the titer of the virus in plaque forming units (PFU) per mL. The titer was 8.4 × 10^9^ PFU/mL.

### 4.4. In Vitro BLI and FLI

Transduced PC-3 cells transfected with pBIT1.2-N [CMV LgBiT were plated with an equal seeding density (50,000 cells/well) in a black 96-well plate (Greiner-Bio-One, Frickenhausen, Germany) in 100 µL of RPMI medium (Sigma, St. Louis, Mo, USA). Transduced Xenograft Human Prostate Cancer cells, the PC339C, expressing LgBiT were plated with an equal seeding density (500,000 cells/well) in 1 mL of growth medium in a black six-well plate with a clear bottom (Sigma-Corning 3506) and the PC346C cells expressing LgBiT were plated with an equal seeding in 100 µL of growth media in a black 96-well plate (Greiner-Bio-One).

Oncolytic adenoviral infection was performed with several dilutions of the initial HAdV-5-DELTA-24-RGD-GFP-T2A-HiBiT viral stock (ranging from 4 to 0.03 MOI). Twenty-four hours after infection the cells were washed with PBS and the bioluminescence signal from wells was measured with IVIS spectrum system (PerkinElmer, Whaltam, MA, USA) 1 min after substrate addition from the Nano-Glo Luciferase Assay System (Promega) with a final concentration of 0.01 mm. The photon flux (ph/s) was collected using open filter binning = medium, field of view = 12.9 × 12.9 cm, f/stop = 1 and either a 30 s or 60 s exposure time. The experiments were performed in triplicates and were repeated three times. Data were analyzed using Living Image 4.3 software (Perkin Elmer) by drawing the appropriate ROI and then plotted using GraphPad Prism 8. Cells were visualized at the microscopy for checking transfection rate using the GFP signal and viability.

### 4.5. Neutralization Assay

The intravenous immunoglobulin G (IvIg) (Sanquin, Leiden, The Netherlands) with a final concentration of 50 mg/mL has been incubated with the oncolytic reporter virus for 1 h and then inoculated to the cells. NanoLuc activity was measured 24 h post infection at the IVIS imager.

### 4.6. In Vivo Bioluminescence

Animal experiments were approved by the Bioethics Committee of Erasmus MC, Rotterdam, The Netherlands under the approved work protocol 17-867-42, covered by the national project license CCD number 2017867. The experiments are performed in accordance with national guidelines and regulations established by the Dutch Experiments on Animal Act (WoD) and by the European Directive on the Protection of Animals used for scientific purpose (2010/63/EU).

BALB/C nude (males) were obtained from Charles River Laboratory (The Netherlands). All mice were provided access to food and water ad libitum and were hosted in the animal facility at the Erasmus MC, Rotterdam, The Netherlands.

For the subcutaneous skin model experiments, eight-week-old nude BALB/C nude (males) were anesthetized using isoflurane and were injected with 2 × 10^6^ PC-3-LgBiT expressing cells prepared in PBS and matrigel (Sigma-Corning) solution (50:50 ratio). After tumor cell implantation (PC-3-LgBiT) and tumor formation, mice were intratumorally (i.t.) injected, from three different sites, with 10 µL (in total 30 µL) of HAdV-5-DELTA-24-RGD-GFP-T2A-HiBiT (viral titer 8.49 × 10^9^ PFU/mL) or PBS. Animals were monitored daily and were euthanized in case of unusual behavior e.g., weight loss > 20% of baseline, rapid tumor growth > 0.5 cm etc.

Mice were monitored over 6 weeks (43 dpi) for virus-treated mice (*n* = 8) and 20 days for control mice (*n* = 4) by non-invasive imaging.

At different time points post intratumoral injection of the virus, 4.2 µmole of formulated HFz [13] in a volume of 480 µL of PBS was injected i.p. in nude mice. Mice were randomly assigned and were kept under (ketamine (25 mg/mL) (Vetalar) and Xylazine (1.7 mg/mL) (Rompun) anesthesia. Anesthetized mice (in groups of three) were placed in a specifically designed imaging box [24] for biosafety level 2 containment purpose.

Series of images were taken from 15 to 20 min after substrate administration using an IVIS Spectrum (Perkin Elmer) with open filter binning = medium, field of view = 12.9 × 12.9 cm, f/stop = 1 and a 3-min exposure time for the imaging of viral infection. At the peak of the bioluminescence signal, regions of interests (ROIs) were used as a tool to analyze the signals.

### 4.7. Immunohistochemistry

To detect the adenovirus hexon proteins, paraffin-embedded sections of the mouse tumors (obtained 43 days post intratumoral viral injection (dpi) for the treated ones and 20 dpi for the control) were deparaffinized and rehydrated with xylene and ethanol according to standard procedures [25]. The sections were then treated with primary polyclonal anti-adenovirus (clone Ab6982; Abcam, Cambridge, MA, USA) and Alexa Fluor 488-labeled goat anti-rabbit (Thermo Fisher-Molecular Probes, Whaltam, MA, USA) antibodies. The nuclei were stained with Hoechst in TBS (1:1000). Stained slides were analyzed by confocal microscopy (Leica Microsystems, Wetzlar, Germany).

### 4.8. Statistical Analysis

Analysis of the bioluminescence output, where more than two groups were compared, was performed using a one-way ANOVA, followed by Tukey’s t-test. All statistics were calculated using GraphPad Prism version 8 for Windows. Data from each animal were presented as means ± SD. The results were statistically significant when *p* < 0.05.

### 4.9. Data Availability

The authors confirm that all relevant data are included in the paper and/or its Appendix A. Other data that support the findings of this study are available from the corresponding author on request.

## Figures and Tables

**Figure 1 ijms-21-05863-f001:**
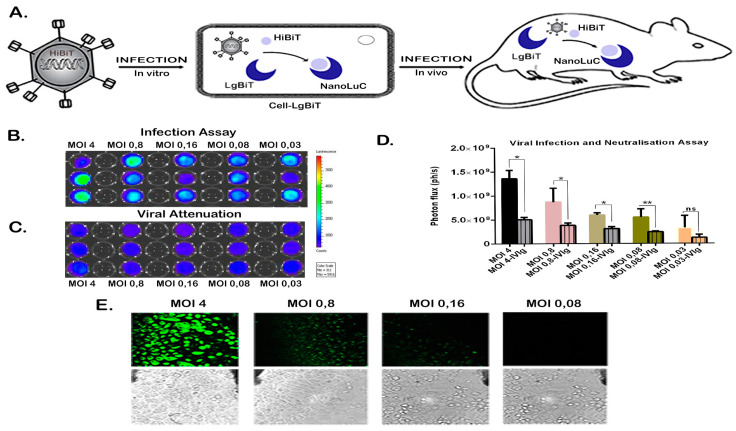
Detection of HiBiT-reporter in vitro. (**A**) Schematic representation of NanoBiT system applied for imaging of viral infection in vivo and in vitro. Bioluminescence in vitro imaging of HiBiT-reporter virus infection by applying the NanoBiT system (**B**,**C**). PC-3-LgBiT cells were infected with: (**B**) HiBiT-reporter virus or (**C**) with neutralized HiBiT-reporter virus by pre-incubation with intravenous immunoglobulin G (IvIg). The HiBiT-reporter virus infection was performed with several dilutions of viral stock (ranging from 4 to 0.03 MOI). The virus-exposed cells were imaged 24 h post infection by addition of substrate from the Nano-Glo Luciferase Assay System with a final concentration of 0.01 mm. (**D**) Signals were quantified with IVIS software. Quantification of detected signal as correlation between detected light and viral infectivity after background subtraction. Results are presented as means +SD. Data are significant different (one-way ANOVA F-value of 30.16) and the signal of infected cells is significantly different from the neutralization assay (* *p*-value < 0.01, ** *p*-value < 0.001) (**E**) Fluorescence microscopy of PC-3-LgBiT cells infected with different HiBiT-GFP-reporter virus dilutions (varying from 4 to 0.03 MOI). For checking the transfection rate using the GFP signal, fluorescence was detected 24 h post infection. Size of the scale bar is 2 µm.

**Figure 2 ijms-21-05863-f002:**
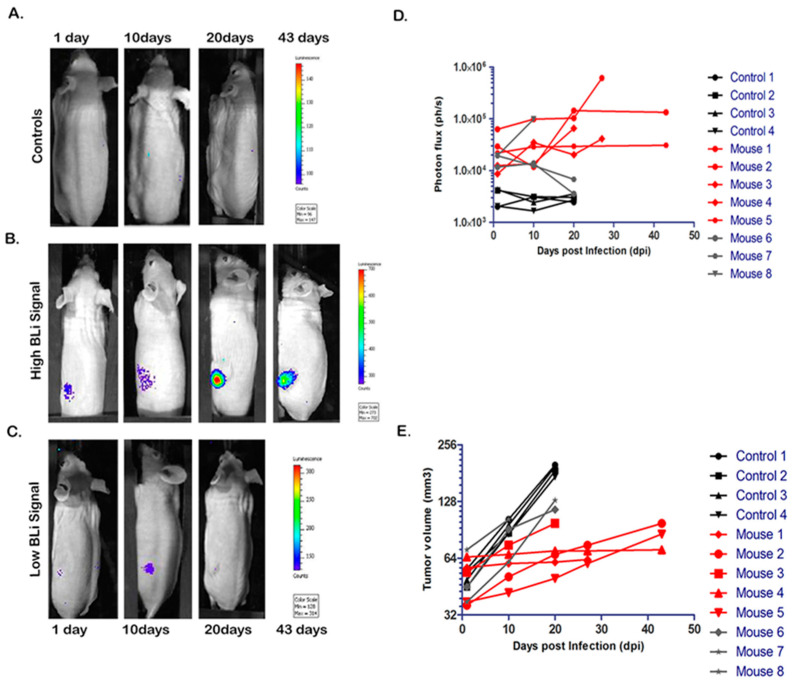
Longitudinal bioluminescence (BLI) imaging of HiBiT-reporter virus infection dynamics in vivo in PC3-LgBiT xenografts. (**A**–**C**) Infection dynamics in nude BALB/C mice infected with 3.0 × 10^7^ PFU of HiBiT-reporter virus. Representative infected mice were imaged at indicated time points by injecting 4.2 µmol of HFz intraperitoneally (i.p.) and monitoring the BLI signal over 43 days for the high BLI signal response group (*n* = 5), 20 days for the low BLI signal response group (*n* = 3) and 20 days for the control group (*n* = 4). (**D**) Signals were quantified at the IVIS software. Results are presented as median +SD (*n* = 4 for the control, *n* = 5 for ‘high BLI signal group’ and *n* = 3 for ‘low BLI signal group’). Light signals were significantly different at 20 days post infection (dpi) (F-value of 4.84) where the ‘high BLI signal group’ was statistically different from the ‘low BLI signal group’ and the control group (*p*-value < 0.05; *p*-value < 0.01). (**E**) Tumor volume changes over time. The volume of tumors treated with oncolytic virus was significantly lower in the group of ‘high BLI signal’ (*n* = 5) when compared to the groups of the control mice (*n* = 4) and the group with ‘low BLI signal’ (*n* = 3) (*p*-value < 0.01) at day 20.

**Figure 3 ijms-21-05863-f003:**
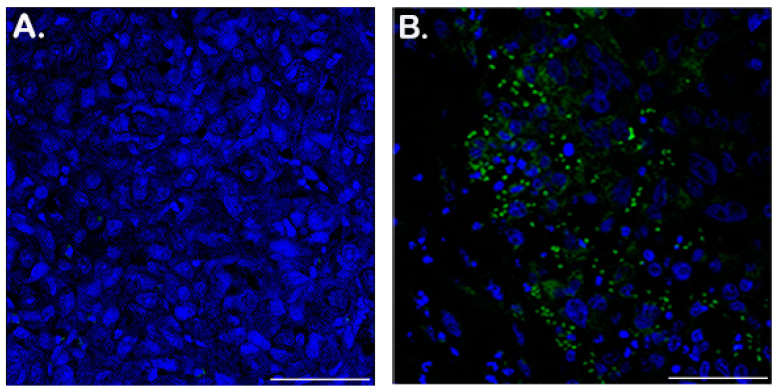
Immunohistochemistry of PC3-LgBiT tumor sections show HiBiT-reporter virus persistence within tumor xenografts 43 days post infection. Virus infected cells assessed via hexon detection in tumor xenografts. Frozen sections of tumors injected with PBS (**A**) or HiBiT-reporter virus (**B**) were stained by immunohistochemistry for adenovirus detection with an anti-hexon antibody and an Alexa Fluor 488-labeled secondary antibody; the sections were counterstained with Hoechst in TBS. (**A**) Representative control section from a tumor injected with PBS. (**B**) A representative section from virus-infected tumors at 43 days post infection. Size of the scale bar is 100 µm.

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
