# Peer review of "NanoBiT System and Hydrofurimazine for Optimized Detection of Viral Infection in Mice—A Novel in Vivo Imaging Platform"

_ijms, 2020, doi:10.3390/ijms21165863_

Round 1

Reviewer 1 Report

Overall the manuscript is sound and the conclusions are supported by the data. However, there are a few key pieces of information that will need to be included to better understand the functionality of the described system. Based on the work presented, I believe these data can be included without necessitating significant new experiments.

Major Issues

Figure 1b – there appears to be variability in infection efficiency or bioluminescent production across the replicate samples. This variability should be explicitly reported and evaluated to determine the inter- and intra-assay coefficients of variation. These data are necessary to understand the functionality of the approach.

Section 2.2 – a crux of this work is the improvement offered by the use of HFz as a substrate. However, little information is provided about the logistics of using HFz with the split luciferase system. It will be necessary to provide information about the duration of the bioluminescent signal post-substrate injection, the kinetics of substrate uptake (at a minimum to include the time between substrate injection and image acquisition), and the effect of different substrate injection routes on bioluminescent emission dynamics (this this can be achieved by highlighting the reason underlying the difference between intratumor vs intraperitoneal injection).

Line 169 – it should be reported if the change in tumor volume was significantly different between the control group and the viral treatment group. Although this data is not the focus of this report, providing these statistics will be helpful for demonstrating the functional capabilities of the reporter system.

Minor Issues

Line 50 – there are only two bioluminescent systems capable of real-time monitoring: bacterial and fungal luciferase. Neither has been used for viral monitoring in vivo. This statement about real-time functionality should be removed, or it should be clarified that bioluminescence can be used for real-time assessment with these systems and they should be properly cited.

Line 101 – grow should be changed to grows.

Line 105 – spurious period should be replaced with a coma.

Line 114 – it is stated that the infection rate was determined, but this information is only presented graphically. It should also be reported numerically for reference.

Figure 1a – this could benefit significantly from being presented in color. It is also difficult to interpret that the small black sphere representing HiBiT is coming from the virus and becoming the larger black sphere that attached to the LgBiT.

Figure 1d – I would recommend presenting the data as paired bars (grouping the bars representing the Infection Assay and the Viral Neutralisation assay together by MOI rather than presenting them as two separate sets of bars. This would make them easier to compare.

Line 157 – this section of the manuscript is written in such a way that it sounds like only a single substrate injection was performed on day one and that only imaging is repeated at the indicated time points. It should be denoted that the subjects were re-injected with substrate and note if any changes needed to be made to the substrate uptake window prior to imaging as a result of changes in animal health, viral load, or tumor growth.

Lines 206 – 210 – HFz was created in a previous work. This section makes it sound like it was developed as part of this effort. This should be revised to make it clear that HFz was selected for this effort, not created as part of this effort.

Line 260 – it would be helpful if the restriction sites identified in the text were underlined in the primer sequences.

Line 281 – superfluous parenthesis should be removed.

Line 282 – this line needs to be edited to improve readability and the superfluous parenthesis should be removed.

Lines 290 – 291 – it would be helpful if the restriction sites identified in the text were underlined in the primer sequences.

Line 301 – fix number formatting.

Line 303 – fix number formatting.

Section 4.5 – the timing of image acquisition post substrate treatment should be noted.

Section 4.6 – the timing of image acquisition post substrate treatment should be noted.

Author Response

Overall the manuscript is sound and the conclusions are supported by the data. However, there are a few key pieces of information that will need to be included to better understand the functionality of the described system. Based on the work presented, I believe these data can be included without necessitating significant new experiments.

We thank the reviewer for his time spent on our manuscript and for giving constructive criticism. We are grateful since the manuscript will be improved.

Major Issues

Figure 1b – there appears to be variability in infection efficiency or bioluminescent production across the replicate samples. This variability should be explicitly reported and evaluated to determine the inter- and intra-assay coefficients of variation. These data are necessary to understand the functionality of the approach.

We thank the reviewer for this comment. Indeed, there is high variability. In general, for enzyme assays, precision is usually <10%; 20 to 50% for in vivo and cell based assays; but can be >300% for virus titer assays. In our case inter-assay CV% are 16% and intra-assay CV% are 80%. To calculate inter-assay variability, we considered the three point were the assay shows linearity MOI (4-0.16). We have changed figure 1b following your suggestion below and added the statistical significance on the data, reported in Figure 1b also in the legend.

Section 2.2 – a crux of this work is the improvement offered by the use of HFz as a substrate. However, little information is provided about the logistics of using HFz with the split luciferase system. It will be necessary to provide information about the duration of the bioluminescent signal post-substrate injection, the kinetics of substrate uptake (at a minimum to include the time between substrate injection and image acquisition), and the effect of different substrate injection routes on bioluminescent emission dynamics (this this can be achieved by highlighting the reason underlying the difference between intratumor vs intraperitoneal injection).

We thank the reviewer for noticing this important aspect that was missing in the manuscript. We noticed the signal was stable from 15 to 20 minutes after injection. We do not know how the signal is in the first 15 minutes since that is the time it takes to transport the animals in the imaging box form Biosafety level 2 animal facility and our imaging facility. However, the effect of different substrate injection routes on bioluminescent emission dynamics are extensively reported in the recently published paper of Yichi Su et. al, we cited. The substrate showed a stable high emission 15-30 minutes after injection when formulated in P-407 so we were confident using that time window. We only injected the substrate i.p. and never intratumorally. Virus was injected intratumorally. We have now included the information about time of imaging in the Material and Methods section, Results and the reference to give a clear information to the reader.

Line 169 – it should be reported if the change in tumor volume was significantly different between the control group and the viral treatment group. Although this data is not the focus of this report, providing these statistics will be helpful for demonstrating the functional capabilities of the reporter system.

We thank the reviewer for this comment. Indeed, we also feel it is important and we have now added the statistical significance in the figure legends. 

Minor Issues

Line 50 – there are only two bioluminescent systems capable of real-time monitoring: bacterial and fungal luciferase. Neither has been used for viral monitoring in vivo. This statement about real-time functionality should be removed, or it should be clarified that bioluminescence can be used for real-time assessment with these systems and they should be properly cited.

We agree with the reviewer and we removed the word real-time which can lead to misinterpretation.

Line 101 – grow should be changed to grows.

We changed in the text.

Line 105 – spurious period should be replaced with a coma.

We have replaced the spurious period with a coma.

Line 114 – it is stated that the infection rate was determined, but this information is only presented graphically. It should also be reported numerically for reference.

We agree with the reviewer that if we mention infection rate then we refer to a number. However, our intention was to show a visual correspondence between GFP positive cells and decreasing MOIs. We have changed the sentence in the text in:” Cells from this assay (Fig.1b and 1c) were imaged to visualize the presence of virus based on the amount of GFP positive cells.”

Figure 1a – this could benefit significantly from being presented in color. It is also difficult to interpret that the small black sphere representing HiBiT is coming from the virus and becoming the larger black sphere that attached to the LgBiT.

We have changed the color of LgBiT so that it is now clear that HiBiT from virus and LgBiT form the NanoLuc luciferase.

Figure 1d – I would recommend presenting the data as paired bars (grouping the bars representing the Infection Assay and the Viral Neutralisation assay together by MOI rather than presenting them as two separate sets of bars. This would make them easier to compare.

We followed the suggestion so now Figure 1d is changed.

Line 157 – this section of the manuscript is written in such a way that it sounds like only a single substrate injection was performed on day one and that only imaging is repeated at the indicated time points. It should be denoted that the subjects were re-injected with substrate and note if any changes needed to be made to the substrate uptake window prior to imaging as a result of changes in animal health, viral load, or tumor growth.

To make this point clear to the reader we have changed the text in: “When imaging was repeated on day 10 post infection (dpi), most tumors still displayed photon emission, after substrate administration, suggesting the presence of the virus in these tumors (Fig.2b and 2c)”. We have clarified the imaging procedures in the Materials and Methods section.

Lines 206 – 210 – HFz was created in a previous work. This section makes it sound like it was developed as part of this effort. This should be revised to make it clear that HFz was selected for this effort, not created as part of this effort.

We thank the reviewer for noticing it and we have now changed the text in: “To enable in vivoimaging of the infection course with our HiBiT-reporter virus, we employed hydrofurimazine, a substrate with improved solubility compared to furimazine [13]. The substrate was formulated in a non-toxic, highly water-soluble excipient, which allows much higher substrate loading and extended light emission in vivoafter a single substrate injection”.

Line 260 – it would be helpful if the restriction sites identified in the text were underlined in the primer sequences.

We now underlined them.

Line 281 – superfluous parenthesis should be removed.

We have removed them.

Line 282 – this line needs to be edited to improve readability and the superfluous parenthesis should be removed.

We have improved readability and removed parenthesis when superfluous.

Lines 290 – 291 – it would be helpful if the restriction sites identified in the text were underlined in the primer sequences.

We have underlined the restriction site.

Line 301 – fix number formatting.

We have now fixed it.

Line 303 – fix number formatting.

We have now fixed it.

Section 4.5 – the timing of image acquisition post substrate treatment should be noted.

We have added this information.

Section 4.6 – the timing of image acquisition post substrate treatment should be noted.

We have added this information.

Reviewer 2 Report

The authors present the system based on split NanoLuc luciferase for BL tracking of viral infection. The obtained results look interesting and valuable.

General remark:

The manuscript looks like it was hastily rewritten for this journal and, consequently, should be organized according to the rules for journal.

I would recommend to introduce the “Results and Discussion” and “Conclusion” sections instead of the “Discussion” section. This will help the authors to avoid unnecessary repetitions.

Minor comments:

There are many typos in the text.

Author Response

We thank the reviewer for the time spent reading our manuscript and the positive remark about the validity of our work.

When we wrote the manuscript we used the word template provided by the journal. There the Results and Discussion sections are two separate sections. Moreover, we feel that keeping discussion separate, indeed, gives us the chance to recapitulate our findings which is an advantage for the reader.

However, we thank the reviewer for noticing the typos so we have carefully re-edited the entire manuscript.